# Association of Energy Expenditure and Efficacy in Metastatic Renal Cell Carcinoma Patients Treated with Nivolumab

**DOI:** 10.3390/cancers14133214

**Published:** 2022-06-30

**Authors:** Johanna Noel, Anne Jouinot, Jérôme Alexandre, Guillaume Ulmann, Marie Bretagne, Zahra Castel-Ajgal, Sixtine De Percin, Clémentine Vaquin-Villeminey, Marie-Pierre Revel, Michael Peyromaure, Pascaline Boudou-Rouquette, Jennifer Arrondeau, Ithar Gataa, Jean-Philippe Durand, François Goldwasser, Olivier Huillard

**Affiliations:** 1Institut du Cancer Paris CARPEM, AP-HP, APHP.Centre, Department of medical oncology, Hôpital Cochin, 75014 Paris, France; johanoel91@hotmail.com (J.N.); jerome.alexandre@aphp.fr (J.A.); marieclaire.bretagne@aphp.fr (M.B.); zahra.castelajgal@curie.fr (Z.C.-A.); sixtine.depercin@aphp.fr (S.D.P.); clementine.villeminey@aphp.fr (C.V.-V.); pascaline.boudou@aphp.fr (P.B.-R.); jennifer.arrondeau@aphp.fr (J.A.); itharg@yahoo.fr (I.G.); jean-philippe.durand@aphp.fr (J.-P.D.); francois.goldwasser@aphp.fr (F.G.); 2Université de Paris, Institut Cochin, INSERM U-1016, CNRS UMR-8104, 75014 Paris, France; anne.jouinot@aphp.fr; 3Endocrinology Department, AP-HP, Hôpital Cochin, 75014 Paris, France; 4Immunomodulatory Therapies Multidisciplinary Study Group (CERTIM), AP-HP, APHP.Centre, Hôpital Cochin, 75014 Paris, France; marie-pierre.revel@aphp.fr; 5Centre de Recherche des Cordeliers, «Equipe labélisée Ligue Contre le Cancer», CNRS SNC 5096, Sorbonne Université, Université de Paris, INSERM, 75005 Paris, France; 6Université de Paris, URP 4466 PRETRAM, F-75006 Paris, France; guillaume.ulmann@aphp.fr; 7Clinical Chemistry Department, AP-HP, Hôpital Cochin, 75014 Paris, France; 8Radiology Department, AP-HP, Hôpital Cochin, 75014 Paris, France; 9Urology Department, AP-HP, Hôpital Cochin, 75014 Paris, France; michael.peyromaure@aphp.fr

**Keywords:** metastatic renal cell carcinoma, nivolumab, anti-PD(L)-1 therapy, hypermetabolism, resting energy expenditure

## Abstract

**Simple Summary:**

Immune checkpoint inhibitors have improved the tumor response and survival of patients with metastatic renal cell carcinoma. However, no predictive factor of response to immunotherapy is currently available. The aim of our study was to evaluate metabolism, assessed with resting energy expenditure as a biomarker of response, in patients with metastatic renal cell carcinoma treated with nivolumab. We found that the 15 patients (29% of the cohort) that had hypermetabolism also had decreased tumor control and tended to have a worse overall survival compared to patients with normal or low metabolism. We conclude that the measurement of resting energy expenditure could help identify patients who will benefit most from immunotherapy in metastatic renal cell carcinoma.

**Abstract:**

Background: Nivolumab improved patients’ survival in metastatic renal cell carcinoma (mRCC). We aimed to evaluate resting energy expenditure (REE) (i.e., patients’ basal metabolism) to predict efficacy. Methods: We conducted a monocentric, observational study of mRCC patients receiving nivolumab between October 2015 and May 2020. REE was measured prior to initiating immunotherapy using indirect calorimetry to determine hypo, normo and hypermetabolism. Primary endpoint was 6-month, progression-free survival (PFS), and secondary endpoints were response rate, PFS and overall survival (OS). Results: Of the 51 consecutive patients, 15 (29%) were hypermetabolic, 24 (47%) normometabolic, and 12 (24%) hypometabolic. The 6-month PFS was 15% for hypermetabolic patients and 65% for non-hypermetabolic patients (*p* < 0.01). In the multivariate analysis, hypermetabolism was the only baseline factor predicting 6-month PFS (OR 9.91, 95%CI [1.62–60.55], *p* = 0.01). Disease progression was noted as the best response in 73% of hypermetabolic patients and 26% of non-hypermetabolic patients (*p* = 0.02). Median PFS was 2.8 and 8.7 months (*p* < 0.01), and median OS was 20.2 and 35.1 months (*p* = 0.13) in the hypermetabolic and non-hypermetabolic groups, respectively. Conclusions: Our study identifies an association between mRCC patients’ energy expenditure and nivolumab efficacy. The measurement of REE by indirect calorimetry in routine practice could help identify patients at risk of nivolumab failure.

## 1. Introduction

Renal cell carcinoma is the 6th and 10th most frequent cancer in men and women, respectively [1]. The prognosis of metastatic renal cell carcinoma (mRCC) has improved during recent decades with Vascular Epithelial Growth Factor Receptor (VEGFR), tyrosine kinase inhibitors (TKI), anti-PD-1 and anti-CTLA-4 immune checkpoint inhibitors (ICI). In the frontline setting, recent studies demonstrated the superiority of sunitinib over ipilimumab plus nivolumab [2], pembrolizumab–axitinib [3], cabozantinib–nivolumab [4] and lenvatinib–pembrolizumab [5], according to the International Metastatic Consortium Database (IMDC) risk subgroups. In the second line after previous TKI treatment, nivolumab as a monotherapy is the only approved ICI [6]. Despite these recent developments, survival remains poor in mRCC, with a survival rate of 5 years at 12%, and primary or secondary resistance to ICI occurring most of the time [7]. As of today, no robust predictive factor of response to ICI exists in mRCC. For example, the PD-L1 status, which is routinely used as a tumor-related biomarker of response in non-small lung cancer (NSCLC), has been inconstantly associated with response to ICI, and is not recommended in clinical practice [8,9,10]. There is an urgent need to identify the patients who will benefit from these treatments and those who are only at risk of toxicity or disease progression.

Cancer-associated weight loss results from negative energy balance, due to decreased food intake, increased resting energy expenditure (REE), or both. REE can be defined as the amount of energy that is spent in one day by a resting individual. Hypermetabolism is defined as an increase in measured REE compared to calculated REE from standard formulas [11] and can affect up to half of cancer patients [12]. It is considered as an early and major contributor to cancer cachexia and was determined as an independent prognostic factor for NSCLC patients’ survival [13]. Furthermore, hypermetabolism is associated with the production of inflammatory cytokines [14], and thus could possibly modulate the response to immunotherapy. We hypothesized that increased REE would result in a reduced ability of the patient to develop an active immune response following ICI infusion. This is because T-cell activation requires metabolic changes to leave the G0 stage and enter the G1 stage of the cell cycle, which includes enhanced glycolysis and is a highly ATP-dependent process [15]. We previously demonstrated that ICI have poor efficacy in hypermetabolic NSCLC patients with a worse prognosis compared to non-hypermetabolic patients [16].

In this context, we decided to evaluate the association between metabolism and efficacy outcomes in mRCC patients treated with nivolumab.

## 2. Patients and Methods

### 2.1. Patients

We conducted a prospective, monocentric, observational study, and thus neither randomized nor blinded, between 1 October 2015 and 31 May 2020 in Hôpital Cochin AP-HP, Paris. We enrolled consecutive patients who participated in the CERTIM (Immunomodulatory Therapies Multidisciplinary Study group) multidisciplinary risk assessment program for immunotherapy in the outpatient unit of the oncology department. This program is proposed to all cancer patients before ICI initiation and aims to provide personalized, supportive and optimized care. During this program, patients benefit from a multidisciplinary evaluation, including a nutritional assessment with a dietitian consultation and indirect calorimetry.

Male and female patients of at least 18 years of age with histologically proven metastatic RCC and indication of nivolumab were eligible. Patients were followed until death or last examination. Follow-up period for the current analysis ended on 2 September 2020.

Written informed consent was obtained for all patients. The study was approved by the Cochin institutional ethical review board (CLEP n°5120518) and procedures were performed according to the revised Declaration of Helsinki.

### 2.2. Data Collection

REE was determined prior to immunotherapy initiation under standard resting conditions, i.e., after 12 h of fasting (overnight), between 8 and 9 a.m. in a thermo-neutral environment by a trained nurse. Before the start of the measurements, patients rested quietly for 15 min while the indirect calorimeter was calibrated and a steady state was obtained [17]. Patients were asked to remain awake for the duration of the measurement. For each patient, oxygen consumption (VO2) was measured for 15 min by indirect calorimetry using a face mask connected to an oxygen analyzer (Fitmate, COSMED, Italy). The calorimeter was calibrated before each measurement. Measured REE (mREE, kcal/d) was determined from VO2 using Weir’s equation [18], and the results were immediately displayed in software attached to the system.

To evaluate the extent of REE alteration compared to healthy individuals, mREE was compared to predicted REE (pREE), calculated with revised Harris and Benedict equations [19]:Males: pREE (kcal/d) = 66.5 + 13.75 × weight (kg) + 500 × height (m) − 6.78 × age
Females: pREE (kcal/d) = 655 + 9.56 × weight + 185 × height − 4.68 × age

Hypermetabolism is defined as mREE over pREE ≥ 110%, normometabolism is defined as mREE/pREE between 90 and 110%, and hypometabolism is defined as mREE/pREE < 90% [11]. In our study, hypermetabolic patients were compared to the other patients (normometabolic and hypometabolic patients).

During the multidisciplinary risk assessment program, multiple clinical and biological measurements were performed. Concomitant medication that could impact the immune response was recorded [20,21]. Risk groups for survival were defined as favorable, intermediate or poor using the IMDC classification [22].

Response to treatment was evaluated by computed tomography at two to four months after treatment initiation, at the discretion of the treating oncologist.

Treatment-related adverse events (TRAE) were defined as all-grade adverse events that could be related to nivolumab, according to the oncologist. A serious adverse event was defined as an adverse event requiring treatment interruption, corticosteroid use and/or hospitalization.

### 2.3. Outcomes

Six-month progression-free survival (6 m PFS), i.e., the proportion of patients that did not have disease progression at 6 months, was the main evaluation criterion since it was determined to be a fair surrogate for the evaluation of efficacy of ICI [23]. Secondary criteria were response rate, PFS and overall survival (OS).

### 2.4. Statistical Analysis

Calculations were performed using R statistical software (version 4.1.0, R Stats, survival and survcomp packages).

Comparisons between groups were performed with the Mann–Whitney–Wilcoxon test for quantitative variables and Fisher’s exact test for qualitative variables.

Based on previous results of the CheckMate-025 study [6], we calculated that we would need to enroll 17 patients in the hypermetabolic group and 33 patients in the normo- and hypometabolic group to show a 50% difference of 6m PFS (80% vs. 40%, respectively) with a two-sided 5% significance level and an 80% statistical power.

Logistic regression was used to test the association of clinical and biological variables with 6-month PFS. Variables not normally distributed were transformed into categorical variables for logistic regression analyses. Significant variables in univariate analyses and clinically relevant analyses such as the IMDC prognostic risk groups were included into multivariable models. Interaction tests revealed no significant subgroup differences. The calibration of multivariate models was checked using the Hosmer–Lemeshow test (adequate calibration with *p* > 0.05 for all models). Finally, likelihood ratio test was used to check the discrimination performance of nested models (*p* < 0.05 for models adding mREE/pREE).

Survival curves were obtained with Kaplan–Meier estimates and compared with log-rank test.

All *p*-values were two-sided, and the level of significance was set at *p* < 0.05.

## 3. Results

### 3.1. Patient Characteristics

Between October 2015 and September 2020, 58 mRCC patients were included in the CERTIM multidisciplinary assessment before nivolumab treatment, and 51 patients were included in the study cohort (Figure 1).

Among them, two patients were lost to follow-up and two had discontinuation of nivolumab before 6 months because of a serious TRAE.

Patient characteristics are presented in Table 1.

Using the standard mREE/pREE ≥ 110% and <90% cut-off, a total of 15 (29%), 24 (47%) and 12 (24%) patients were classified as hypermetabolic, normometabolic and hypometabolic, respectively. Patients were predominantly men, median age was 63 years, and performance status (PS) was good. Most patients (60%) had received one previous TKI regimen. Eight patients received frontline nivolumab due to TKI ineligibility or sarcomatoid features on the tumor [24]. Characteristics were well-balanced between the hypermetabolic and non-hypermetabolic groups. Neutrophil-to-lymphocyte ratio (NLR) at four to six weeks of treatment did not differ between the two groups (*p* = 0.74 and 0.67, respectively; data not shown).

Two patients had a papillary renal cell carcinoma. They were in the non-hypermetabolic group, and nivolumab was used as second-line treatment in both patients.

### 3.2. Efficacy

The median follow-up was 19.2 months. The 6m PFS was 15% in hypermetabolic patients and 65% in non-hypermetabolic patients (*p* < 0.01). In the univariate analysis, increased REE and the absence of TRAE were unfavorably associated with 6m PFS (Table 2).

In multivariate analysis, these two parameters were associated with 6m PFS, with an OR of 9.91 for hypermetabolism (95%CI [1.62–60.55], *p* = 0.01) (Table 3).

Disease control rate, defined as the proportion of patients with complete response, partial response, or stable disease, was statistically higher in the non-hypermetabolic group, with 74% and 27% in the non-hypermetabolic and hypermetabolic groups, respectively (*p* < 0.01, Table 4). The two patients with a papillary histology subtype had stable disease as the best response, and the duration of response was 4 and 5 months.

At data cut-off, 13 of 14 patients (93%) in the hypermetabolic group had disease progression compared to 25 of 35 (71%) in the non-hypermetabolic group. Median PFS was 2.8 months in the hypermetabolic group and 8.7 months in the non-hypermetabolic group (*p* < 0.01, Figure 2).

Twenty-seven deaths were observed: 11 in the hypermetabolic group (73%) and 16 in the non-hypermetabolic group (44%). Median OS was 20.2 months (range 1.3–45) and 35.1 months (range 1–48.4) in the hypermetabolic and non-hypermetabolic groups, respectively (*p* = 0.13, Figure 3).

### 3.3. Toxicity

All-grade TRAE occurred in 4 (27%) and 19 (53%) of the hypermetabolic and non-hypermetabolic patients, respectively (*p* = 0.13). The main TRAE were cutaneous lesions (33%), dysthyroidism (8%) and pneumonitis (8%). Among patients presenting a TRAE, seven presented a serious adverse event: five patients in the non-hypermetablic group (two interstitial pneumonitis, three serious cutaneous reaction); and two patients in the hypermetabolic group (one pneumonitis and one myositis with cardiac insufficiency) (*p* = 0.57); with no death associated with the reported treatment toxicity.

## 4. Discussion

To our best knowledge, this is the first study evaluating the association of resting energy expenditure and efficacy of ICI in mRCC. We found that non-hypermetabolic patients receiving nivolumab had better outcomes than hypermetabolic patients, with a 6m PFS of 65% and 15%, respectively. The objective response rate was also two-fold higher in the non-hypermetabolic group than in the hypermetabolic group, and hypermetabolism resulted in a significantly decreased disease control rate as compared to non-hypermetabolism.

This report is the first study including a measurement of REE in mRCC patients treated with ICI. The diagnosis of hypermetabolism can be obtained in routine practice in the outpatient setting using indirect calorimetry, with a result immediately available for the physician. REE reflects the total amount of energy used by an individual per day. It depends, for example, on weight, sex and age, and its value varies from one person to another. A vast array of phenomena can increase REE in cancer patients. The energetic demand of the tumor itself, changes in inflammation and body composition, will result in increased REE. It is estimated that up to half of cancer patients may be hypermetabolic [12,16,25]. We confirmed in this study the high prevalence of hypermetabolism with 29% of patients being hypermetabolic. Cancer cachexia is a multifactorial syndrome with an ongoing loss of skeletal muscle mass that cannot be fully reversed by conventional nutritional support and leads to progressive functional impairment [26]. Hypermetabolism has been correlated with clinical and biological markers of cancer cachexia such as weight loss, PS ≥ 2, CRP ≥ 10 mg/L and associated with reduced survival [12,13]. Hypermetabolism is an early feature of cachexia. Therefore, indirect calorimetry should be encouraged and performed early and systematically for need and risk assessments of cancer patients, prior to initiation of cancer therapies.

Nivolumab was approved for the treatment of mRCC following the pivotal CheckMate-025 study [6]. Patients’ characteristics in both studies are comparable, with a high proportion of males, 16% in the unfavorable IMDC risk group, and patients that received mainly one or two prior lines of treatment before nivolumab. Efficacy outcomes for the overall population of our study were similar to the nivolumab group in the CheckMate-025 study, with 45% and 40% of 6m PFS, and 33% and 25% of objective response rate, respectively. Furthermore, primary resistance to nivolumab was reported in 39% of our overall study population (73% of hypermetabolic patients and 26% of non-hypermetabolic patients), and in 35% of the nivolumab cohort of the CheckMate-025 study. Interestingly, survival surrogates were better in the non-hypermetabolic group of our study than in the nivolumab group in the CheckMate-025 study: median PFS was 8.7 months in our study group versus 4.6 months in the CheckMate-025 study; and median OS was 35 months versus 25 months.

In our study, we did not find any predictive marker of nivolumab efficacy other than hypermetabolism and occurrence of TRAE. In mRCC, no easily available biomarker of response to immunotherapy has been identified, but multiple studies are currently trying to evaluate the tumor, tumor microenvironment and host-related biomarkers. Promising biomarkers related to the tumor include molecular signatures, such as in the BIONIKK trial [27]. Host-related factors offer new potential biomarkers in mRCC such as the gut microbiome [28] and systemic inflammation reflected by NLR [29,30]. Finally, the IMDC risk group has been identified as a predictive factor of response to nivolumab plus ipilimumab [2]. The predictive value of TRAE has also been studied [31,32] but it cannot help as a predictive marker before initiation of treatment. Interestingly, we previously reported an association between patients’ hypermetabolism and acute toxicity under nivolumab [33].

The main limitations of our study are its monocentric nature and its limited sample size. In our cohort, eight patients received nivolumab as frontline treatment, which may not have a comparable efficacy as in the post TKI setting [34,35,36]. The value of hypermetabolism as a biomarker of response under TKI treatment or ICI combination with ICI or TKI is currently unknown. We decided to include all mRCC patients in our cohort, including two patients with papillary histology subtype since we believe that metabolism is a host-related, histology-independent marker. Although the efficacy of ICI in papillary renal carcinoma may differ from that of ICI in clear cell carcinoma, these two non-hypermetabolic patients had stable disease as a best response.

This study also has strengths. Despite a limited number of patients, there is a strong indication that hypermetabolism should be investigated as a marker of efficacy in larger cohorts. Our population is representative of the real-life setting in mRCC, with patient characteristics and outcomes consistent with the literature and similar to that of the pivotal CheckMate-025 study [22,37].

It can be highlighted that, in our study, hypermetabolism is not only associated with a poor PFS but also with a reduced tumor response rate, suggesting a direct link between whole-body energy and the possibility of a pharmacodynamic effect of ICI. Generating an immune response is a considerable bioenergetic challenge [15]. In order to respond appropriately to the tumors, T-cells have to metabolically switch between resting and proliferative states and actively acquire metabolic substrates from their environment to meet these energy demands [38]. Since hypermetabolism indicates that increased energy is spent by the patient in resting conditions, to maintain vital homeostasis, we hypothesized that hypermetabolic patients might have less remaining energy to fuel lymphocytes, and therefore may experience less lymphocytes-mediated tumor responses [39,40].

In conclusion, indirect calorimetry allows for an easy and reproducible measure of REE, and our findings point toward its interest as a marker of efficacy for immunotherapy in mRCC patients. Future studies will need to explore this biomarker in the frontline setting in different ICI combinations and determine the efficacy of strategies to reverse patients’ hypermetabolism.

## Figures and Tables

**Figure 1 cancers-14-03214-f001:**
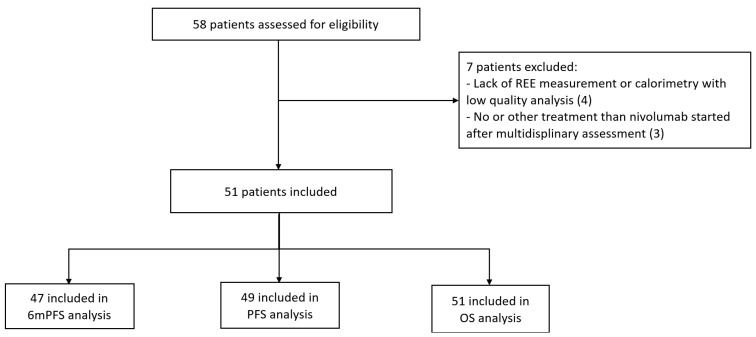
Flow chart. REE: resting energy expenditure; PFS: progression-free survival; 6 m PFS: 6-month progression-free survival; OS: overall survival.

**Figure 2 cancers-14-03214-f002:**
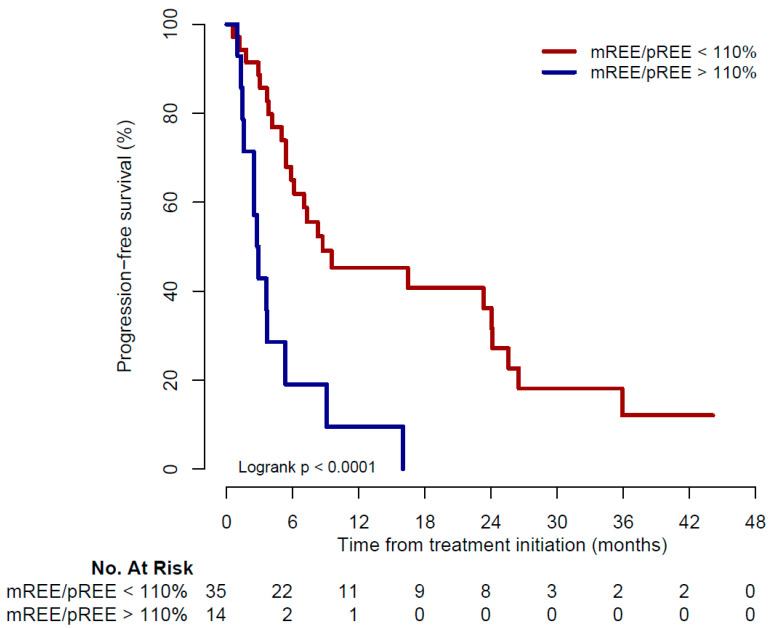
Kaplan–Meier curve for progression-free survival. mREE: measured resting energy expenditure; pREE: predicted resting energy expenditure.

**Figure 3 cancers-14-03214-f003:**
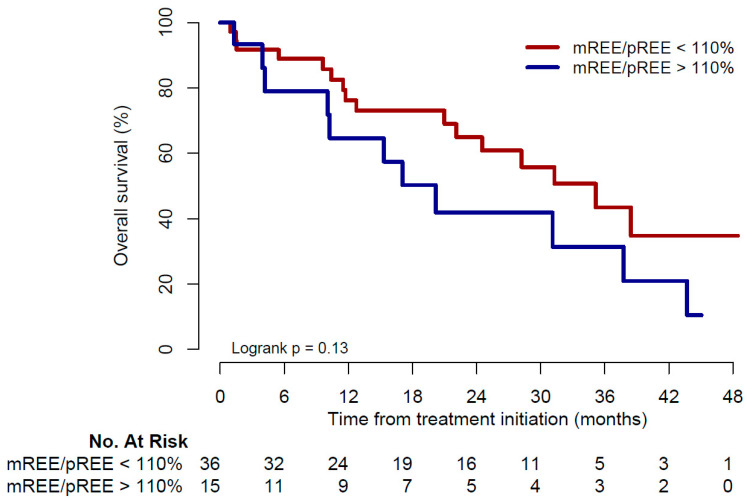
Kaplan–Meier curve for overall survival. mREE: measured resting energy expenditure; pREE: predicted resting energy expenditure.

**Table 1 cancers-14-03214-t001:** Patient characteristics.

	Total (*n* = 51)	Hypermetabolic Group (*n* = 15)	Non-Hypermetabolic Group (*n* = 36)	*p*-Value
**Age (years), median (range)**	63 (42–93)	60 (42–83)	66 (51–93)	0.08
**Female sex, n (%)**	39 (76)	12 (80)	27 (75)	1
**Concomitant medication**				
**Beta blockers, n (%)**	15 (29)	4 (27)	11 (31)	
Corticosteroids	6 (12)	1 (7)	5 (14)	
Insulin	6 (12)	2 (13)	4 (11)	
Antibiotics	1 (2)	0	1 (3)	
**Histology, n (%)**				
Clear cell	49 (96)	15 (100)	34 (94)	
Papillary type 2	2 (4)	0	2 (6)	
**Other histology component, n (%)**				0.77
None	25 (49)	6 (40)	19 (53)	
Rhabdoid	1 (2)	0	1 (3)	
Sarcomatoid	6 (12)	2 (13)	4 (11)	
NA	19 (37)	7 (47)	12 (33)	
**Nephrectomy, n (%)**				0.67
No	7 (14)	3 (20)	4 (11)	
Yes	44 (86)	12 (80)	32 (89)	
**IMDC risk group, n (%)**				0.38
Poor	8 (16)	4 (27)	4 (11)	
Intermediate	25 (49)	7 (47)		
Favorable	18 (35)	4 (27)	14 (39)	
**Metastatic at diagnosis, n (%)**				0.53
No	32 (63)	8 (53)	24 (67)	
Yes	19 (37)	7 (47)	12 (33)	
**Site of metastasis, n (%)**				
Lung	38 (75)	9 (60)	29 (81)	0.16 *
Lymph node	26 (51)	8 (53)	18 (50)	1
Bone	22 (43)	6 (40)	16 (44)	1
Liver	11 (22)	4 (27)	7 (19)	0.71
Adrenal gland	10 (20)	3 (20)	7 (19)	1
Pancreas	6 (12)	2 (13)	4 (11)	1
Brain	2 (4)	0	2 (6)	1
**Number of previous lines of treatment, n (%)**				0.06
0	8 (16)	1 (7)	7 (19)	
1	31 (60)	8 (53)	23 (64)	
2	9 (18)	4 (27)	5 (14)	
>2	3 (6)	2 (13)	1 (3)	
**Performance status, n (%)**				0.47
0	4 (8)	2 (13)	2 (6)	
1	30 (59)	9 (60)	21 (58)	
2	15 (29)	3 (20)	12 (33)	
3	2 (4)	1 (7)	1 (3)	
**Weight loss, n (%)**				0.45
<5%	41 (80)	11 (73)	30 (83)	
≥5%	10 (20)	4 (27)	6 (17)	
**Body mass index (kg/m^2^), n (%)**				0.21
≥25	34 (67)	8 (53)	26 (72)	
<25	17 (33)	7 (47)	10 (28)	
**Treatment-related adverse event, n (%)**				0.13
Yes	23 (45)	4 (27)	19 (53)	
No	28 (55)	11 (73)	17 (47)	
**CRP (mg/L), n (%)**				0.54
≥10	29 (57)	10 (67)	19 (53)	
<10	22 (43)	5 (33)	17 (47)	
**Albumin (g/L), n (%)**				1
≤35	8 (16)	2 (13)	6 (17)	
>35	42 (82)	12 (80)	30 (83)	
NA	1 (2)	1 (7)	0	
**NLR increase, n (%)**				0.74
Yes	25 (49)	8 (53)	17 (47)	
No	21 (41)	5 (33)	16 (44)	
NA	5 (10)	2 (13)	3 (8)	

IMDC: International Metastatic Database Consortium; mREE: measured resting energy expenditure; pREE: predicted resting energy expenditure; CRP: C-reactive protein; NLR: neutrophil-to-lymphocyte ratio. *: all metastatic sites are compared to each other.

**Table 2 cancers-14-03214-t002:** Univariate analysis of 6-month progression-free survival.

	OR [95% CI]	*p*-Value
**Age** /year increase	0.98 [0.93–1.04]	0.52
**Sex** Male vs. Female	0.76 [0.18–3.28]	0.39
**IMDC risk group** Intermediate vs. favorable Poor vs. favorable	0.5 [0.07–3.67] 0.75 [0.06–8.83]	0.50 0.82
**Metastasis at diagnosis** Yes vs. No	1.04 [0.31–3.52]	0.94
**Nephrectomy** Yes vs. No	1.58 [0.24–10.44]	0.64
**Number of previous lines of treatment** ≥2 vs. 0–1	2.08 [0.51–8.4]	0.31
**Number of mestastic sites** ≥3 vs. 0–2	0.84 [0.26–2.71]	0.77
**Tumor total size** /cm increase	1.22 [0.98–1.52]	0.08
**Performans status** ≥2 vs. 0–1	1.82 [0.52–6.37]	0.35
**Weight loss in the last 6 months** ≥5% vs. <5%	1.32 [0.31–5.71]	0.71
**Body Mass Index** ≥25 kg/m^2^ vs. <25 Kg/m^2^	0.98 [0.88–1.09]	0.71
**Treatment-related adverse events** Yes vs. No	0.23 [0.07–0.8]	**0.02**
**mREE/pREE ratio** ≥110% vs. <110%	9.62 [1.82–50.89]	**<0.01**
**CRP** /point increase	1.02 [0.99–1.04]	0.23
**Albumin** ≤35 g/L vs. >35 g/L	1 [0.18–5.56]	1
**NLR at baseline** /point increase	1.15 [0.95–1.4]	0.15
**NLR increase** ≥25% vs. <25%	1.14 [0.32–4.08]	0.84
**Creatinin over cystatin ratio** /point increase	0.99 [0.96–1.02]	0.52

IMDC: International Metastatic Database Consortium; mREE: measured resting energy expenditure; pREE: predicted resting energy expenditure; CRP: C-reactive protein; NLR: neutrophil-to-lymphocytes ratio.

**Table 3 cancers-14-03214-t003:** Multivariate analysis for 6-month progression-free survival.

	HR [95% CI]	*p*-Value
**mREE/pREE ratio** ≥110% vs. <110%	9.91 [1.62–60.55]	**0.01**
**Treatment-related adverse events** Yes vs. No	0.24 [0.06–1]	**0.049**
**IMDC intermediate risk group** Yes vs. No	0.78 [0.17–3.59]	0.75
**IMDC poor risk group** Yes vs. No	0.47 [0.04–5.28]	0.54

mREE: measured resting energy expenditure; pREE: predicted resting energy expenditure; IMDC: International Metastatic Database Consortium.

**Table 4 cancers-14-03214-t004:** Efficacy of nivolumab.

	Total (*n* = 51)	Hypermetabolic Group (*n* = 15)	Non-Hypermetabolic Group (n = 36)	*p*-Value
**PFS rate at 6 months**, n (%)	23 (45)	2 (15)	22 (65)	**<0.01**
**Best response**, n (%)				**0.02**
Complete response	5 (10)	1 (7)	4 (11)	
Partial response	12 (24)	2 (13))	10 (29))	
Stable disease	13 (25)	1 (7)	12 (34))	
Progressive disease	20 (39)	11 (73)	9 (26)	
NA	1 (2)			
**Objective response rate**, n (%)	17 (33)	3 (20)	14 (40)	0.52
**Disease control rate**, n (%)	30 (59)	4 (27)	26 (74)	**<0.01**
**Number of infusions**, median (range)	6 (1–27)	5 (1–24)	8 (1–27)	**0.04**

PFS: Progression-Free Survival; objective response rate: patients who had a partial or complete response to treatment; disease control rate: patients who had a stable disease, partial response or complete response to treatment.

## Data Availability

The datasets generated and/or analyzed during the current study are available from the corresponding author on reasonable request.

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
