# Peer review of "Association of Energy Expenditure and Efficacy in Metastatic Renal Cell Carcinoma Patients Treated with Nivolumab"

_cancers, 2022, doi:10.3390/cancers14133214_

Round 1

Reviewer 1 Report

In their study Johanna Noel et al. analyzed association of energy expenditure and efficacy in mRCC patients treated with Nivolumab. The study addressed a potentially interesting issue. Renal cell carcinoma (especially metastatic form) is therapy resistant. Nivolumab was approved for mRCC therapy in 2015, so it is relatively new drug and information about all new factors influencing therapy is important. The authors proved that patients with increased REE have worse response for Nivolumab. It is important finding which could potentially help oncologist to choose right therapeutic strategy for hypermetabolic patients.

As the authors mentioned the main limitation of this study is number of patients. The study was performed on 47 patients (for 6mPFS) the group of hypermetabolic patients consist only 15 cases. On that small group authors found only two factors which influences 6 month free survival (mREE/pREE ratio and TRAE). Other predictive factors associated with Nivolumab response in cancer patients: CRP (10.18632/oncotarget.21602), NLR at base line and NLR increase ( doi: 10.3389/fonc.2021.746976. ) were not statistically relevant. This fact  suggest that group was too small for detecting predictive factors other than mREE/pREE ratio and TRAE.

I recommend the study of Johanna Noel et al. for publication in the Cancers

Author Response

We thank the reviewer for his / her review and this positive comment.

Reviewer 2 Report

The manuscript analyzes the expenditure of energy in metastatic renal cell carcinomas previously treated with nivolumab. The manuscript is well organized and the results are clearly exposed.

However, in my view, the manuscript would be significantly better if the two papillary carcinomas are excluded from the study. In fact, 2 papillary carcinomas add nothing to the results obtained in 49 clear cell renal cell carcinomas. Besides, papillary renal cell carcinomas are very differen tumors, with very different genesis and clinical course. What the authors say wih 51 cases can be said equally with 49. A work considering only one type of noplasm is more robust. Accordingly, I would add in the title clear cell renal cell carcinoma instead of renal cell carcinoma.

Author Response

We thank the reviewer for his / her review.

We understand the reviewer main comment since papillary RCC are rarely studied along clear cell RCC and the efficacy of immune checkpoint inhibitors may differ in these subtypes.

Nevertheless, we decided to include papillary tumors in our cohort because immune checkpoint inhibitors are currently prescribed in the second line setting for these tumors like for clear cell RCC and we think that patients’ metabolism is a histology-independent, host-related biomarker (like performance status). In our opinion the results are in line with this hypothesis. The two patients with papillary RCC were non-hypermetabolic, nivolumab was used as second line of treatment, they had stable disease as best response, the duration of response was 4 and 5 months and the survival 10 and 24 months, respectively. Since these two patients being in the non-hypermetabolic group had no upfront progression but had progressed at 6 months, we think that they do not represent a bias in the results. We agree with the reviewer that this is information should have been included in the manuscript and it was therefore added in the results and in the discussion sections.